

# Seasonal sea ice forecast skills and predictability of the KMA's GloSea5

Byoung Woong An[1], Sang Min Lee[1], Pil-Hun Chang[1], KiRyong Kang[1], and Yoon Jae Kim[1]

[1]Earth System Research Division, National Institute of Meteorological Sciences, Seogwipo, Jeju, Korea

**Correspondence:** Byoung Woong An (bwan@korea.kr)

**Abstract.** Ensemble sea ice forecasts of the Arctic Ocean conducted with the Korea Meteorological Administration's coupled global seasonal forecast system (GloSea5) is verified. To investigate the temporal and spatial characteristics of the seasonal projection of Arctic sea ice extent and thickness, a set of ensemble potential predictability is assessed. It shows significance for all lead months except anomalous around East Siberian Sea, Chukchi Sea and Beaufort Sea during summer months. However, during the rapidly thawing and freezing season, initial states lose its predictability and increase uncertainties in the prediction. The probability skill metrics show the summer sea ice prediction which strongly depends on the sea ice thickness interacting with the accuracy of the snow depth. We found the forecast skill is determined primarily by the timing of sea ice drift (i.e., Beaufort Gyre and Transpolar drift) and sea ice formation by freshwater flux in the East Siberian Sea. Therefore, capturing the sea ice thickness state effectively is the key process for skillful estimation of Arctic sea ice. In spite of the uncertainties in atmospheric conditions, this system provides skillful Arctic seasonal sea ice extent predictions up to six months.

*Copyright statement.* TEXT

## 1 Introduction

Arctic ocean could be a region where ocean and atmospheric processes interact more sensitively and so susceptible to sea ice prediction. In particular, exchanges of heat, mass and momentum across the air-sea interface are enhanced significantly in the Arctic ocean by large air-sea temperature differences and high wind speeds. Moreover, the timing and speed of seasonal sea ice variation modulates these processes and turn to contributes to uncertainty concerning the sea ice spatial and temporal variations. Therefore the seasonal sea-ice prediction is operated by the coupled global climate model. Seasonal forecasts are forecasts with timescales of a few weeks up to a few months and the sources of predictability depends on its timescale. In addition, seasonal forecast aims to estimate probabilities and provides the range of values which is most likely to occur during the next season as anomalies in long-term climatic conditions to a certain extent due to the complex and stochastic nature of the atmospheric circulation. This research focused on the predictability of the Arctic sea ice on a seasonal scale.

The climate variability at seasonal to interannual time scales and its effects on the atmosphere were investigated by Alexander et al. (2004) and Bhatt et al. (2008). Recently, predicting seasonal to interannual variability of the sea ice cover has drew





attention of many scientific research group (e.g., Drobot et al., 2007; Lindsay et al., 2008, 2012), and many works have been done to explore the inherent predictability of Arctic sea ice (Döscher et al., 2010; Mikolajewicz et al., 2005). Lindsay et al. (2008) found that much of the predictive information in the ice-ocean system is lost for lead times greater than about three months by using a linear empirical model with atmospheric circulation indices, ocean temperature, and sea ice data.

Predicting sea ice cover in the Arctic could be possible by using cumulative observations of sea ice and advances in modeling sea ice at seasonal to interannual time scales. Satellite observations of sea ice extent and area are recorded now for more than 40 years. Nearly four decades of consistent satellite observations have reported declines in sea ice extent (the total area of the Arctic with at least 15 % sea ice concentration) significantly for each month of the year (Serreze and Stroeve, 2015; Stroeve and Notz, 2015). Changes are largest in summer and smallest in winter, with September trends (month with the lowest sea

ice cover; 1979 to 2017) of -83,000 $km^2 yr^{-1}$ (-13.0 % per decade), and -41,000 $km^2 yr^{-1}$ (-2.7 % per decade) for March (month with the greatest sea ice cover; Onarheim et al. (2018). Spatially, the regions of summer ice loss are dominated by changes in the East Siberian Sea (explains 22 % of the September trend), with large declines also observed in the Beaufort, Chukchi, Laptev and Kara seas (Onarheim and Årthun, 2017). Winter ice loss is dominated by reductions within the Barents Sea, responsible for 27 % of the pan-Arctic March sea ice trends (Onarheim and Årthun, 2017). Reconstructions of the sea

ice cover back to 1850 using earlier satellite observations, ship and aircraft observations, ice charts, and whaling records show that Arctic ice loss over the past two decades is likely unprecedented in at least 150 years (Walsh et al., 2017). Holland et al. (2008) and Goosse et al. (2009) found the variability of sea ice area increased significantly over the 21st century, which suggests predictability might decline.

       Seasonal forecasts are currently under utilized for prevention, adaptation and prediction by the public and economic sectors.

As the Arctic sea ice thinners, the areal coverage and its thickness are more variable. Hence predicting subseasonal to seasonal variability of the sea ice cover is important for local Inuit communities for their hunting (Fox, 2003) and for shipping industries needed advanced information of the opening of the northwest passage and northern sea route (Hassol, 2004; Stephenson et al., 2011) and for improving mid-latitude weather forecast skill (Jung et al., 2014). Consequently this matter has led to the development of a number of numerical sea ice seasonal prediction systems based on general circulation models (e.g., Sigmond

et al., 2013; Chevallier et al., 2013; Wang et al., 2013; Guemas et al., 2014), as well as systems based on empirical methods (e.g., Schröder et al., 2014; Stroeve et al., 2014). However, these seasonal forecast systems are limited in predicting summer sea ice and diagnosing the source of forecast errors (Day et al., 2014). This is partly because the existing skill and reliability of Seasonal forecast in Arctic is low and varies considerably depending on the geographical area, the time of the year and the climate variable (Bruno and Suraje, 2015). This is also due to insufficient representation of relevant physical processes in the

numerical model and inadequate knowledge of the initial state of key variables e.g., sea ice thickness and subsurface ocean state variables (Day et al., 2014).

       Lemke et al. (1980) found that the main patterns of Arctic sea ice area variability showed a decorrelation time scale (or loss of persistence) of 2 to 5 months. This time scale was found to be associated with heat storage in the marginal ice zone mixed layer. Flato (1995) calculated a relaxation time of 7 years for total Arctic sea ice volume in a model forced by monthly observed

air temperature anomalies. Using a single column stochastic model of sea ice, Bitz et al. (1996) reported that relaxation time




scales fell from 15 to 6 years for Arctic sea ice volume when accounting for ice export through Fram Strait. Lindsay et al. (2008) noted that the importance of properly predicting thickness to improve predictions of sea ice area. Due to the lack of consistent observations of ice thickness, Kwok and Rothrock (2009) point to a role for climate modeling to extend predictions of ice area beyond the initial 2 to 5 month decorrelation time scale found by Lemke et al. (1980).

In this study we assess the predictability of Arctic sea ice in an advanced climate model, GloSea5 (MacLachlan et al., 2014) based on the ensemble prediction method. This ensemble prediction produces multiple estimates of variables based on current conditions and past observations and it provides probabilistic forecast information rather than deterministic (Day, 1985). Probability and uncertainty are derived from the distribution of the predicted values. We evaluate mechanisms for persistence with the model output by comparing their character with observations. Provided the climate models successfully
characterise such mechanisms that give rise to sea ice persistence, these models could be used to make subseasonal to seasonal forecasts.

## 2   Model description and data analysis

### 2.1   Model description

Global seasonal prediction system (GloSea5) is developed by the UK Met Office (MacLachlan et al., 2014) and KMA has
adopted the framework for the operational forecasting and joint monthly-seasonal forecasting with UK Met Office since 2013. This system utilises a Stochastic Kinetic Energy Backscatter scheme (SKEB) to generate spread between ensemble members initialized from the same analysis. Maximum forecast length is 240 days and run every day and make ensemble once a week. The number of perturbed ensemble members is four per day. Hindcast covers for the period from 1991 to 2010 and three ensemble members are initialized on fixed calendar dates, i.e., 1st, 9th, 17th and 25th.
GloSea5 is coupled with an ocean model, NEMO version 3.4 (Madec, 2008; Megann et al., 2014) with a 0.25 degree horizontal resolution and 75 vertical levels and also coupled with a multi-ice thickness-category sea ice model, CICE version 4.1 (Hunke and Lipscomb, 2010) which are initialized from UK Met Office Ocean Analysis (NEMOVAR, Mogensen et al., 2009, 2012; Waters et al., 2014) with frequency of 3 hourly atmosphere-ocean coupling and assimilated sea ice concentration and no assimilation of sea ice thickness (Blockley et al., 2013). The NEMO global ocean configuration uses the ORCA025
grid (with 28 km horizontal grid spacing at the equator reducing to 7 km at high southern latitudes, and around 10 km in the Arctic Ocean) and is based on the configuration developed by Mercator Ocean. The vertical coordinate system is based on geopotential levels using the DRAKKAR 75 level set which provides an increased near surface resolution (including 1 m surface layers to help resolve shallow mixed layers and potentially capture diurnal variability) without compromising resolution at depth. Partial cell thicknesses at the ocean floor allow a better representation of ocean topography and in combination with
an energy and enstrophy conserving momentum advection scheme and a free slip lateral boundary condition improve the mesoscale circulation and in particular the simulation of western boundary currents. Due to constraints when running coupled to the atmosphere model, Unified Model, GloSea5 uses Semtner's zero layer thermodynamic model with a single layer of both ice and snow. Ice dynamics are calculated using the elastic-viscous-plastic (EVP) scheme of (Hunke and Dukowics, 2002). For





technical reasons the GloSea5 system still uses a constant freezing temperature unlike the forced model where the salinity is considered.

KMA run the Unified Model as the atmosphere component of the GloSea5 and its horizontal resolution is N216 ($0.83° \times 0.56°$, about 60 km in mid latitudes). Number of the model levels is 85 (top of model is 85 km) with terrain-following hybrid height coordinates. In addition Global Coupled 2.0 (GloSea5-GC2) uses the Joint UK Land Environment Simulator (JULES). The JULES model is described in Best et al. (2011). In GloSea5-GC2 the soil moisture is initialised from a seasonally varying climatology. This climatology was derived from a JULES re-analysis using Global Land 3.0 and this re-analysis was completed on a 0.5 degree grid and interpolated to the model resolution ($0.83 \times 0.56$ degrees). Snow is initialised from the analysis. For the hindcast the snow field is interpolated from $0.75 \times 0.75$ degrees (ERA-Interim) to the GloSea5-GC2 grid and only snow mass is initialized.

The data assimilation uses NEMOVAR (Mogensen et al. (2009); Mogensen et al. (2012); Waters et al. (2014)), a variational (3D-var) scheme developed for use with NEMO and further tuned for the $1/4°$ global model. Key features of NEMOVAR are the multivariate relationships which are specified through a linearised balance operator and the use of an implicit diffusion operator to model background error correlations. Operationally there is 24 hour data assimilation cycles performed each day where observations are assimilated using a 24 hour window and increments are applied to the model with an incremental analysis update (IAU) step. The coupled GloSea5 system must be initialised from the T+00h analysis. Observations for assimilation include satellite SST data (AVHRR data supplied by the GHRSST project), in situ SSTs from moored buoys, drifting buoys and ships, sea level anomaly observations from Jason2 and CryoSat-2, subsurface temperature and salinity profiles (from Argo, moored buoys, etc.) and sea ice concentration (SSMIS data provided by OSI-SAF (EUMETSAT , 2017) as a daily gridded product).

## 2.2 Data analysis

Monthly output from the large ensemble is used for this study. Each ensemble member is initialized from the same sea ice, land, and ocean conditions with the same coupled model. The initial conditions of the atmosphere are different for each ensemble member and correspond to the state of the atmosphere.

For verification of the forecast quality, we use the monthly Met Office Hadley Centre Sea Ice and Sea Surface Temperature (HadISST) dataset downloaded from https://www.metoffice.gov.uk/hadobs/hadisst/ (Rayner et al., 2003) for calculating Sea ice area and extent. In addition, weekly Arctic sea-ice thickness derived from CryoSat-2 and SMOS using an optimal interpolation scheme is used (Ricker et al., 2017; Yang et al., 2014; Kaleschke et al., 2012; Laxon et al., 2013). Unfortunately this data does not cover from May to September, therefore the verification in this study is limited for only data available period.

We calculate the total Arctic sea ice area by integrating the ice concentration over Northern Hemisphere grid cells. Sea ice extent is calculated as the sum of grid cell areas whenever the sea ice concentration is greater than 15 %. Both area and extent are analyzed because they offer slightly different information about the system. Sea ice area is a useful index of the Arctic climate (Flato, 1995) and arguably more relevant than extent when considering the influence of sea ice on surface albedo and heat fluxes. However, sea ice extent in observations is arguably of better quality than area (Parkinson and Cavalieri, 2008).





To provide and to identify the forecast quality of the GloSea5 and to give more reliable prediction, we use statistical ensemble verification methods and these methods are used to generate ensemble probability distributions with quality characterized by skill (accuracy: the forecasts close to the observed) and spread (variability: the forecast appropriately represent the uncertainty).

## 3 Results

First, we examine how ensemble members diverge over time. Figure 1 shows ensemble spread error of the forecast with 42 ensembles for Arctic sea ice extent of monthly possible extent from January 2016 to September 2017. This ensemble spread error represents the average difference between the individual ensemble forecasts of a quantity and the ensemble mean forecast of the quantity. One month ensemble forecast consists of 42 separate forecasts made by the same model and all activated from the different starting time. The starting conditions for each member of the ensemble are slightly different. The differences between these ensemble members tend to grow as the forecasts progress, that is as the forecast lead time increases. Among all of the monthly forecast ensemble spread error range, August to October have a wider forecast uncertainty compared to other months. In particular the spread error range in the forecasts varies sharply from July to September and down from September to December as for thaw/freeze period. As the skill is weak while there is still discernible spread in the inter-quartile range (25 ∼ 75 % exceedance levels) illustrated by the height of the spread error range box. Note that uncertainties for September predictions are clearly distinguished from other months. These results also imply uncertainties in considering the summer atmospheric variability over Arctic ocean. This ensemble spread error may represent the true variability (uncertainty) of the nature.

We also present box plots of Ensemble mean bias between the ensemble mean in month initialized in each month and its corresponding data of HadISST in Fig. 2 (a). This figure shows the system has a negative bias i.e., the sea ice concentration prediction is always under-estimated compared to the measured data and its bias range is also wider in summer months. Because of the Glosea5 has negative bias as shown in Fig. 2 (a), so for each month, 90th percentile of the ensemble value is evaluated by its root mean square error (RMSE) magnitude and ranges (including all lead times) initialised in each month against HadISST (see Fig. 2 (b)). This RMSE value gives the skill of how closely each month's error ranges. The RMSE of August and September sea ice extent is significantly higher than the other month and so very little skill in that period. We also calculate the Nash-Sutcliffe model efficiency (NE) of the monthly sea ice extent forecast. It is defined as in Eq. 1(Nash and Sutcliffe, 1970) and represents the degree of covariance with the observed value. Figure 2 (c) shows the NE tends to degrade as time goes to summer months. The median values of all of the variables presented in Fig. 2 (c) are respectable. However, the 25th percentile for August and September is quite low. Negative NE values indicate that the model predictions perform more poorly than the observed mean value. There is little variability of performance in the different month, with winter months in both RMSE and NEs being characterized by better performance, and the summer months being characterized by worse





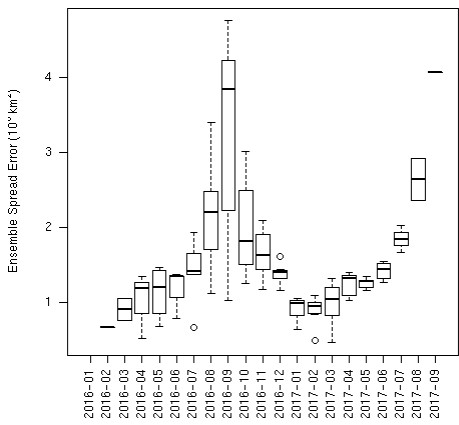

**Figure 1.** Ensemble spread error range of monthly Arctic sea ice extent forecast with 42 ensembles from January 2016 to February 2017. The unit of the error is $10^6 km^2$. In each boxplot, the bottom, middle and top of the box are the 25th, 50th and 75th percentiles, and the bottom and top whiskers are the 10th and 90th percentiles.

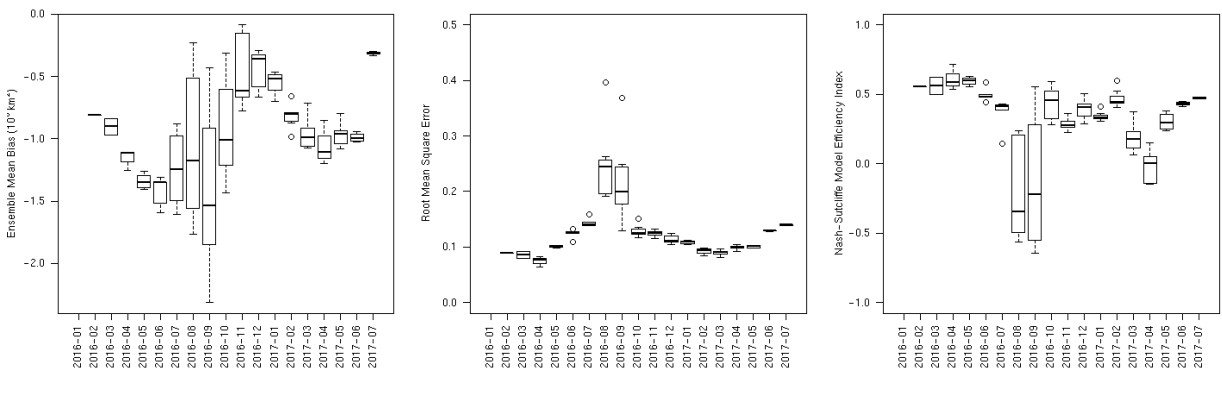

(a) Ensemble Mean Bias  (b) Root Mean Square Error  (c) Nash-Sutcliffe Index

**Figure 2.** Box plots of Ensemble mean bias of Arctic sea ice extent of monthly forecast with 42 ensembles from January 2016 to February 2017 (a), Root Mean Square Error range (b) and Nash-Sutcliffe modelling efficiency range (c) between 90th percentile of the ensembles and HadISST's values.





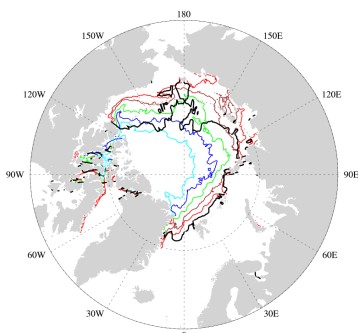
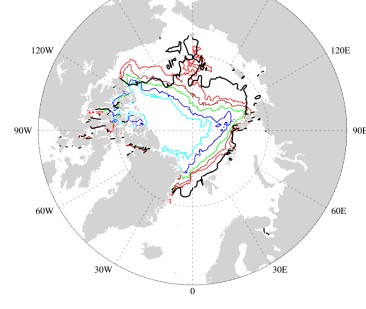

(a) March forecast September sea ice concentration.

(b) June forecast September sea ice concentration.

**Figure 3.** Spatial pattern of sea ice concentration contours (15 % isoline) represented sea ice edge. The colour gives the percentiles of the concentration in 10th (light blue), 25th (blue), 50th (green), 75th (brown) and 90th (red) percentile of ensemble value ranges in September 2016. The black contour line represents HadISST's 15 % sea ice concentration. September Arctic sea ice extent forecasts using March ice concentration data (Fig. 3 (a)) and June ice concentration data (Fig. 3 (b)), respectively.

performance (Fig. 2). This degree of accuracy would likely result in the model error as sea ice melting processes vary across time step.

$$NE = 1 - \frac{\sum (Q_{obs} - Q_{mod})^2}{\sum (Q_{obs} - \bar{Q_{obs}})^2} \tag{1}$$

where $Q_{obs}$ is the mean of observed value, and $Q_{mod}$ is modeled value.

In order to understand the regional predictability of Arctic sea ice, we exam the spatial and temporal differences of sea ice variables i.e., sea ice concentration, thickness and snow depth in percentiles.

    Figure 3 (a) and 3 (b) show March and June forecast September sea ice distribution represent the 15 % sea ice concentration (SIC) isolines of ensemble value ranges percentiles, respectively. The sea ice conditions for the Beaufort and Chukchi seas show a large spread of sea ice edges, indicating large ensemble spreads and high variability within the model. The reference data

(black line) is based on the satellite-based sea ice i.e., HadISST. For September sea ice extent forecast, generally we can see the large variability in the Beaufort Sea where a typically important area for September sea ice forecasts, East Siberian sea and small variability in the Laptev and Barents Seas. September Arctic sea ice extent forecasts using June ice concentration data (Fig. 3 (b)) showed slightly different spatial pattern and was generally underestimated than using March data. That is, more than 75 % percentile is close to the observed value. The main reasons for uncertainties in the Beaufort Sea and East Siberian

sea are probably due to the inaccuracy of the initial sea ice thickness condition (Day et al., 2014).

    Figures 4 (a) and 4 (b) show the edge of the first year ice defined as 30 cm of March and October sea ice 30 cm thickness predicted in September and April, respectively. All the predictions have no significant variance between percentiles except





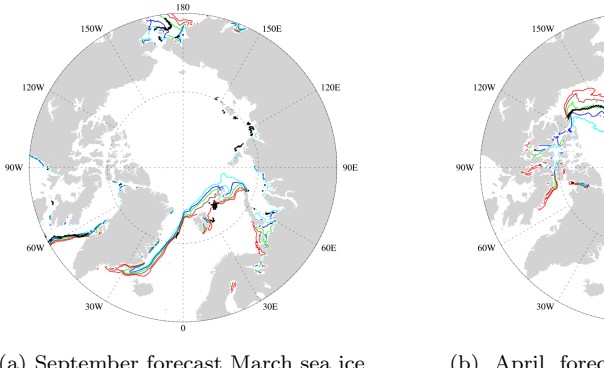

(a) September forecast March sea ice thickness (30 cm).

(b) April forecast October sea ice thickness (30 cm).

**Figure 4.** Spatial pattern of sea ice thickness of the first year ice edge and multi year ice edge. The colour gives the percentiles in 10th (light blue), 25th (blue), 50th (green), 75th (brown) and 90th (red) of ensemble value ranges of 30 cm ice thickness. The black line represents CryoSat-2's 30 cm sea ice thickness isoline each.

October which has wide uncertainty and difference from the observed values. The predicted sea ice thickness percentiles resemble that thick sea ice exists around the Canadian Archipelago and extends to the center of the Arctic and to the East Siberian Sea with the thickness diminishing. The presence of thick sea ice along the coast of East Siberian during the winter is also different from that of observed (not shown here). This can be interpreted as the effect of fresh water flow along the coast

of East Siberian Sea. This is similar to those of the 15 % SIC pattern (Fig. 3 (a)), but there is a wide difference between SIC in the Sea of Chukchi and SIT in the East Siberian Sea.

Figure 5 (a) and (b) show March snow depth forecasted in September and February, respectively. All ensemble predictions have significantly under estimated for the September forecast with wide differences between percentiles, but no significant variance for the February forecast March snow depth. The predicted snow percentiles resemble that of sea ice thickness dis-

tribution around the Canadian Archipelago and extends to the center of the Arctic. The presence of snow connected to the Chukchi Sea and East Siberian Sea can be interpreted as the effect of the snow transition to bare sea ice, but this snow depth is not shown in the measured data. This non-realistic snow depth can be inferred as the effects of the atmospheric model forecast error which related to wind and humidity. This operational prediction systems show some skill in predicting winter snow depth, but diagnosing the source of forecast errors is problematic. Such forecast errors may be due to incomplete knowledge of the

initial state of key variables such as sea ice thickness which is not well observed (Mathiot et al., 2012). Because the Arctic Ocean is almost completely surrounded by the land, there is not as much moisture available and snow fall is relatively low. Therefore the position of atmospheric pressures and sea ice condition affect heat flux and snow precipitation in Arctic ocean.





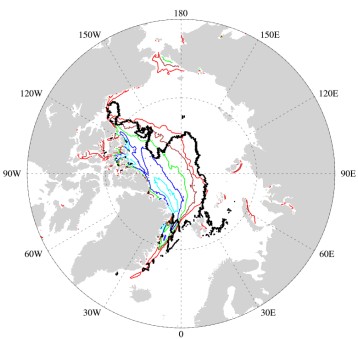
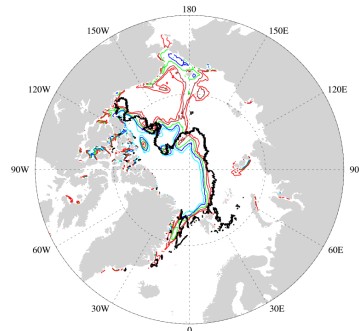

(a) September forecast March snow depth.

(b) February forecast March snow depth.

**Figure 5.** Spatial pattern of snow depth contours (30 cm isoline) represented the height of the snow transition to bare sea ice edge. The colour gives the percentiles of the snow depth (30 cm) in 10th (light blue), 25th (blue), 50th (green), 75th (brown) and 90th (red) of ensemble value ranges in March 2017. The black contour line represents CryoSat-2's 30 cm snow depth.

## 3.1 Probabilistic forecast verification

In previous section, we have assessed the GloSea5's forecast skill by conventional summary statistical methods, such as RM-SEs and bias (mean error), and found the skill depends on percentiles used in assessment and also the bias behaviour depends on seasons. Those methods would be useful for the forecast accuracy assessment, but would not contain the proper information

to apply the verification statistics i.e., unable to evaluate the probabilistic nature of ensemble forecasts because they only evaluate whether a forecast is either right or wrong. Instead of producing a single deterministic forecast, there are many different methods commonly used for verification of probabilistic forecasts e.g., ranked probability score (RPS), ranked probability skill score (RPSS), generalised discrimination score for ensemble, and reliability. These diagnostic verification has been developed by Murphy and Winkler (Murphy and Winkler, 1992, 1987). Murphy et al. (1989) used probability of precipitation and max-

imum temperature forecasts. Wilks (1995) described the methods in detail and discussed their application to meteorological forecasts and Wilks (2000) used diagnostic verification in a study of Climate Prediction Center average temperature and total precipitation long-range forecasts over the United States. However, these methods have not been applied extensively to sea ice forecasts. We compute generalised discrimination score for ensemble and the continuous ranked probability score (Wilks, 2006, CRPS) with the unbiased (fair) version of the CRPS from the SpecsVerification package (Siegert et al., 2017) written in

R (http://cran.r-project.org). A probabilistic forecast, such as ensemble prediction, can be evaluated as a set of values between 1 and 0, and they are therefore neither right nor wrong (Wilks, 1995). The observation will have a value of either 0 (did not occur) or 1 (did occur) (Murphy and Winkler, 1992). Each of the above scores ($S$) may be converted into a skill score ($SS$) by





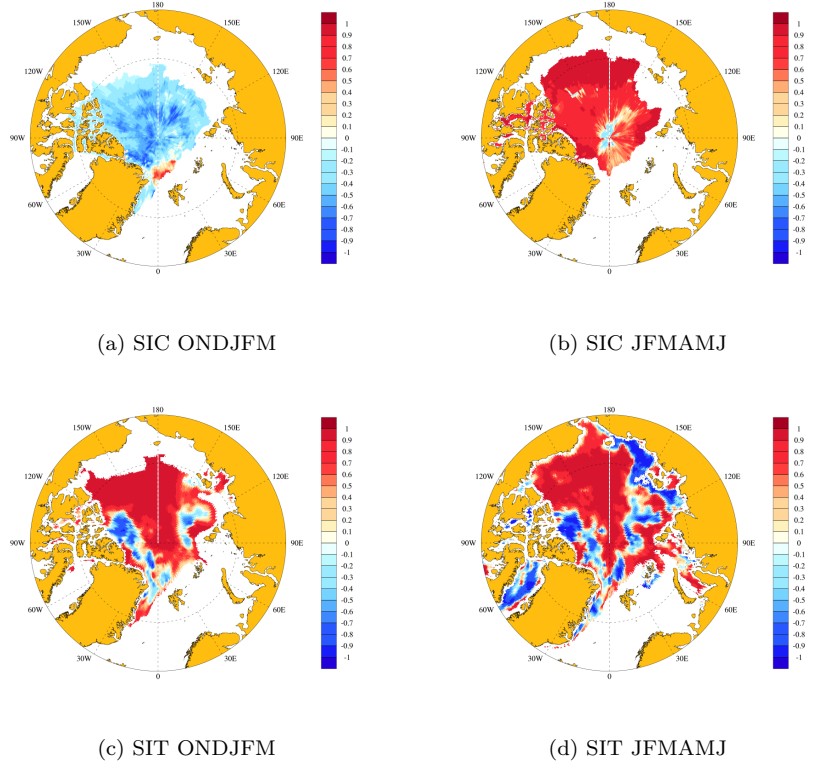

<table>
(a) SIC ONDJFM      (b) SIC JFMAMJ
(c) SIT ONDJFM      (d) SIT JFMAMJ
</table>

**Figure 6.** Two biannual temporal correlation for Arctic Ocean between 2016-2017. The four maps on the top and the bottom correspond to seasonal sea ice concentration and thickness, respectively. Red colour indicates positive correlation and blue colour indicates negative correlation.

comparison with the score evaluated for a reference forecast, $S_{ref}$ :

$$SS = 1 - \frac{(S - S_{ref})}{(S_{pref} - S_{ref})}. \tag{2}$$

For all the scoring rules above the perfect score, $S_{perf}$ is zero and the skill score can be expressed as

$$SS = 1 - \frac{S}{S_{ref}}, -\infty < SS \leq 1. \tag{3}$$

5     This verification aims to account for field spatial structure, provide information on error in physical terms and account for uncertainties in location and timing. This purpose of the verification is to look to find the distribution, location, and pattern errors of the forecast. First we calculate the correlation between the ensemble forecasts and the observation which is influenced by large errors than smaller errors for the verification of sea ice concentration and sea ice thickness for the same period of forecast.



The strength of the correlation between the forecast and observations represents the strength of their joint distribution and it shows whether the spatial features have coherent spatial structure or not. For the evaluation of this technique, biannual temporal correlation was computed for the whole forecasting period i.e., six months in each initial month. Extended seasonal (biannual) sea ice concentration and thickness forecast for each initial month was obtained by averaging over all ensemble members, ONDJFM (October to March forecasted in September) and JFMAMJ (January to June foracasted in December). Correlation was computed on these biannual data sets for each grid point. Figure 6 presents the result of correlation analysis against the verifying observations from HadISST and AWI CryoSat-2 data (Grosfeld et al., 2016). Areas covered in red colour indicate positive relationship and suggest better skill compared to climatology and areas covered in blue indicate worse skill than climatology.

We find that the ensemble mean biannual forecasts for 2016-2017 is not correlated well with the verifying observations of sea ice concentration over most of the central Arctic in autumn and winter season, but also forecasts do not skillfully represent the sea ice thickness over Canadian archipelagos and Eastern Siberian Sea in winter and spring season.

Based on the maps in Fig. 6, the forecast skill for biannual sea ice concentration depends on initial forecasting month, i.e., forecasted in September shows negative correlation in whole Arctic Ocean, on the other hand, the forecast skill forecasted in December is positive and the skill is the whole Arctic Ocean mostly. Contrarily, the biannual sea ice thickness skill depends on regions and slightly depends on initial forecasting month compared to sea ice concentration. This is because of differences in sea ice thickness initial condition and its high seasonal variability by the sea ice drift. Therefore sea ice thickness is hard to observe and to forecast. The skill is mostly in the Beaufort Gyre and Transpolar drift and rarely in the Eastern Siberian Sea and the Canadian archipelagos. It is interesting to note that the area where the models are skillful do not overlap very much. This suggests that the mechanisms leading to the skill in forecast might be different between sea ice concentration and sea ice thickness. The overall lack of significance in the correlation in the Eastern Siberian Sea and the Canadian archipelagos could be because of the low predictability of Fist Year Ice and the initial Multi Year Ice thickness.

Application of generalized discrimination score for ensembles and CRPSS can provide the forecaster and forecast user insight into forecast skill for predicting different flow conditions. This information is valuable to a decision maker when deciding whether or not to rely on the forecast and also can use this information to indicate confidence in the predictions and to adjust them accordingly.

Discrimination measures how well the forecasts discriminate between events and non-events. Ideally, the distribution of forecasts in situations where the forecast event occurs should differ from the corresponding distribution in situations where the event does not occur. We compute the generalised discrimination score for ensemble forecasts with continuous observations as described in Weigel and Mason (2011). Extended seasonal (biannual) forecasts produced by GloSea5 forecasts of sea ice concentration and thickness, averaged over months (October to March and January to June), have been used. Verification is gridpointwise against data from HadISST and CryoSat-2. The resulting skill maps are shown in Fig. 7. In Fig. 7, the observations have not been binned at all i.e., "raw" observation values have been used, and the score has been calculated with Kendall's rank correlation coefficient (Sheshkin, 2007) for continuous observations. The skill patterns of the sea ice concentration are showing that the seasonal predictability of sea ice concentration is higher for January-June forecasting than for October-March,



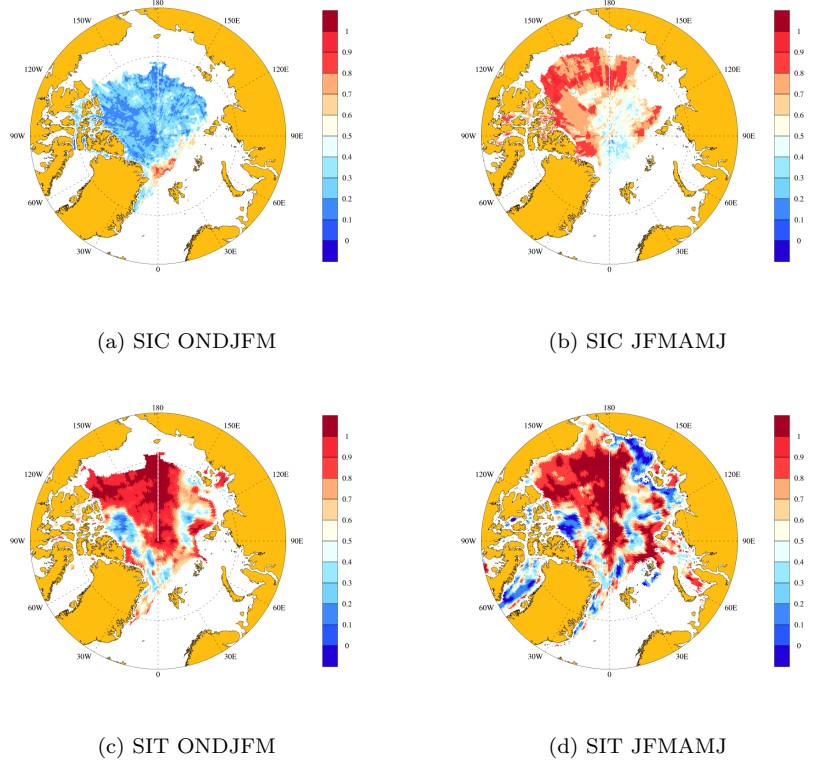

|  |  |
| :---: | :---: |
| (a) SIC ONDJFM | (b) SIC JFMAMJ |
| (c) SIT ONDJFM | (d) SIT JFMAMJ |

**Figure 7.** The generalised discrimination score for GloSea5 forecasts biannual (ONDJFM and JFMAMJ) sea ice concentration (top) and sea ice thickness (bottom) in the Arctic Ocean. The observations have not been binned at all i.e., continuous observations.

especially along the Canadian archipelagos to the East Siberian Sea. Explain this in more detail, the skill average over the Arctic sea ice concentration for forecasting October to March is less than 0.5 for continuous observations, implying that less than 50 % of the cases, the forecasts are able to correctly discriminate. Whereas, the skill average over the Arctic sea ice concentration for December to June is more than 80 % along the Canadian archipelagos to the East Siberian Sea. However, in sea ice

5   thickness cases, the discrimination score patterns are consistent with each other and showing that predictability of the sea ice thickness is highest in the Beaufort Gyre and the Transpolar drift area and lowest in the Canadian archipelagos and Siberian shelf. Discrimination measures whether forecasts differ when their corresponding observations differ or not, but it does not show the quality of the forecasts. Instead, it shows the usability of the forecasts after calibration and post-processing (Weigel and Mason, 2011).

10   Next, we compute the continuous ranked probability skill score (CRPSS). This score operates using absolute values of the forecast and observation, thus no conversion to probabilities is required. We use the unbiased (fair) version of the CRPSS. CRPSS is based on Continuous ranked probability score (CRPS) which is related to the ranked probability score, but compares





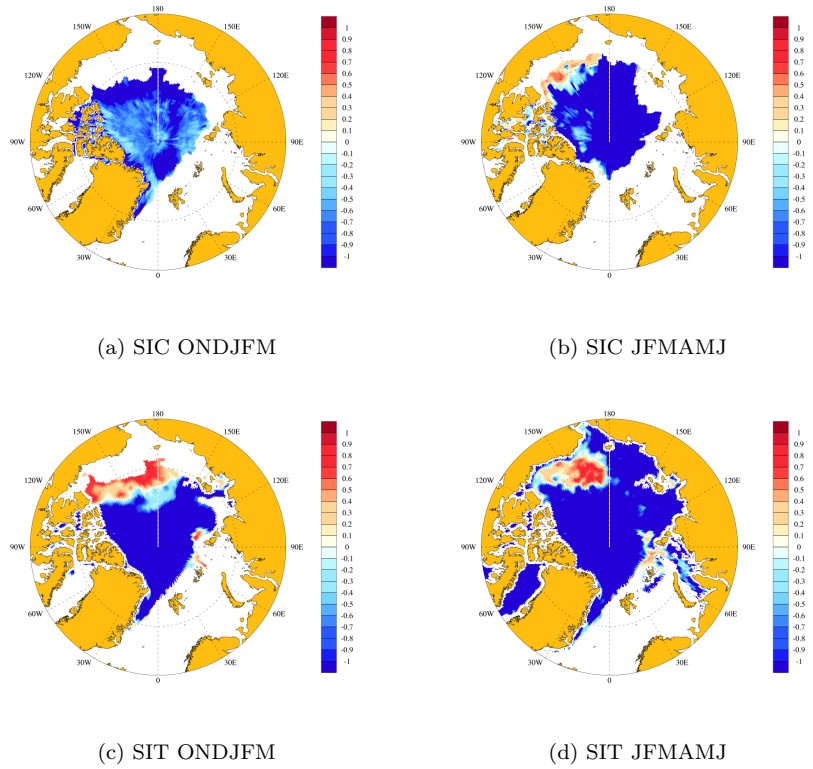

(a) SIC ONDJFM          (b) SIC JFMAMJ

(c) SIT ONDJFM          (d) SIT JFMAMJ

**Figure 8.** CRPSS for biannual (ONDJFM and JFMAMJ) sea ice concentration (top) and sea ice thickness (bottom) between 2016 - 2017 in the Arctic Ocean. Red colour indicates that the skill of the forecast is higher than that of climatology and blue colour indicates that the skill of the forecast is worse than that of climatology.

a full distribution, where both are represented as cumulative distribution functions (cdfs). The equation for calculation of the CRPS is,

$$CRPS(forecast) = \frac{1}{ncases} \sum_{i=1}^{ncases} \int_{x=-\inf}^{x=\inf} (F_i^f(x) - F_i^o(x))^2 dx \tag{4}$$

where $F_i^f(x)$ is the forecast probability cdf for the ith forecast case and $F_i^o$ is the observation, expressed as a cdf.

5  CRPSS measures the relationship between ensemble forecast and observation, and it takes into account the full probability distribution obtained from the ensemble members and compares it with the verifying observation. It can also be interpreted as probabilistic generalization of the mean absolute error. The result of CRPSS on the assessment of biannual sea ice concentration forecast over Arctic Ocean for forecasting October to March show very limited correspondence between the probability distribution of ensemble members and the verifying observations. However, there is some positive and significant skill in sea



ice thickness, which is limited over the Beaufort Gyre and Chukchi Sea area (see Fig. 8). The CRPSS skill for the other areas remains very low. Explain it more detail, CRPSS is indicated as the percentage of the probabilistic forecast error e.g., CRPSS value 0.2 represents that the probabilistic forecast error is 20 % less than the climatological forecast error. Negative values (represented in blue colour) imply that the skill of estimated forecast probabilities is worse than the use of climatological

frequencies as forecast and positive values (represented in red colour) of CRPSS indicate that the model is better than climatological probabilities. White areas on sea surface represents scores that are lower than -1 and showing particularly worse relationship between the ensemble distribution and the observation.

Consistent with the case of correlation of sea ice concentration, CRPSS for forecasting January to June is better predictability compared to that of for forecasting October to March. This implies that seasonal concentration predictability is higher for spring

than for autumn season in general. However, inconsistent with the case of correlation of sea ice thickness, CRPSS for autumn is better compared to that of spring. This suggests that seasonal thickness predictability is roughly higher for autumn than for spring season.

So far we compared the GloSea5 ensemble verifications created by different methods. The maps show different spatial variation structures depending on the method. The maps also show that correlation and CRPSS produce spatial patterns that

are more consistent across different forecast months, therefore, seem to be more robust predictors. In addition, significantly high correlation but fairly low CRPSS skill in sea ice forecast indicates that although the skill to predict the seasonal sea ice concentration and thickness over Arctic ocean is high in many regions, predicting the entire region of the seasonal sea ice concentration and thickness forecasts is still in challenge. This requires re-calibration of the models to reduce the negative skill and requires reconstruction of the initial sea ice thickness that contribute to seasonal sea ice thickness anomalies.

**4    Conclusions**

This paper examined the ability of GloSea5 to predict seasonal variability of Arctic sea ice. Using the KMA GloSea5 predictions start dates of 01, 09, 17, 25 each month from January 2016 to February 2017, forecasting skills of the Arctic sea ice have been analysed. The September outlook from the June initialisation is relatively unbiased. However, large biases are developed when initialised in March. The First Year Ice and snow depth forecasts are well performed, but generally overestimated.

Sources of error are mainly related open water bias especially in marginal ice zone (Chukchi Sea and East Siberian Sea). For sea ice concentration, GloSea5 system seems to have a good predictability in general except for summer months. In particular, forecasting for September has not been so accurate due to large biases that develop when initializing with July and August. For other seasons, using 75 % percentile of the Ensemble predictions provides excellent forecasting performance.

In addition, predictions of Arctic sea ice is assessed based on GloSea5 system the predictability characteristics of Arctic sea

ice conditions on biannual timescales with monthly average ensemble sea ice concentration and thickness for a six month lead time. We assess the biannual sea ice concentration and thickness forecasts against observations dataset of sea ice concentration (HadISST) and sea ice thickness (CryoSat-2) over Arctic Ocean. In this study, we show that the potential for seasonal sea ice forecast skill is particularly high for forecasting January to June especially in the Beaufort Gyre and the Transpolar drift re-





gions. We use the spatial temporal correlation coefficient, the generalised discrimination score and the fair CRPSS as measures to assess the skill of biannual climate forecast. The correlation analysis between the forecast and observed sea ice show that the predictive skill is associated with the lead times except for the summer season. This is because the model is capable of predicting the Arctic sea ice, but there is no obvious linkage between the areal extent and lead time for the summer sea ice pre-

diction. Assessing the divergence among the ensemble members reveals that sea ice exhibits potential predictability for winter conditions, but shows little potential predictability for the spring season. Sea ice forecasts indicate that sea ice concentration in a higher variable sea ice regime generally shows higher potential predictability for a longer period of time. Significantly high correlation but fairly low CRPSS skill indicates that although the skill to predict the seasonal sea ice concentration and thickness over Arctic ocean is high in many regions, predicting the full distribution of the seasonal sea ice concentration and

thickness is still in challenge. This requires re-calibration of the models to reduce the negative skill. In general, the ensemble spread gives an estimate of relevant uncertainty which lies both in the initial state of the forecast and in the forecast model. Therefore the system predictability depends on initial conditions and model physics. The predictability characteristics of the system may change with large-scale ice loss and seasonally dependent mechanisms i.e., the effects of sea ice drift and melt. If the sea ice extent varies primarily by the sea ice drift, we would simply see the multi year ice in the Beaufort and Chukchi Seas

get pushed together. Ogi et al. (2008) showed summer sea level pressure on September ice extent induced Ekman transport of sea ice and resulted to net ice convergence. Holland and Stroeve (2011) also showed the relation between August sea level pressure and September ice extent and concluded that high sea level pressure leads low sea ice.

The results of the assessment show that the skill for seasonal forecasting exists in the Arctic. However, the skill for seasonal sea ice concentration for forecasting October to March is lower than for forecasting January to June. The results also show

less thick ice and a less dispersed multiyear ice cover. The Arctic sea ice prediction on seasonal time scales is closely related to energy balance components of the seasonal sea ice anomalies. However anomalies in sea ice thickness are less correlated with sea ice concentration variability. The accuracy of sea ice prediction is sensitive to sea ice thickness conditions, so it can be improved by applying improved initial sea ice thickness condition. In addition, the higher quality initial data from the atmospheric-sea ice-ocean will allow more accurate predictions of sea ice up to one year after, which will also help predict the

relationship between Arctic sea ice changes and mid-latitudes weather patterns.

*Code availability.*  The model configurations described here are based on the Unified Model (Atmosphere), NEMO (Ocean), CICE (Sea-ice) and JULES (Land Surface). Due to intellectual property right restrictions, we cannot provide either the source code or documentation papers for the UM or JULES. However, the UM is available for use outside the Met Office through a licensing agreement. A number of research organizations and national meteorological services use the UM in collaboration with the Met Office to undertake basic atmospheric process

research, produce forecasts, develop the UM code and build and evaluate earth system models. More information on the use of the UM, as well as the opportunities and support available for collaboration can be found on the UM Partnership page. For further information on how to apply for a licence see http://www.metoffice.gov.uk/research/modelling-systems/unified-model. The ocean model code is available from the NEMO webpage (http://forge.ipsl.jussieu.fr/nemo/wiki/Users) under the CeCILL free software license (http://www.cecill.info/). On registering, individuals can access the code using the open-source Subversion software (http://subversion.apache.org/). The user man-

ual for the NEMO modelling code is available online (https://www.nemo-ocean.eu/bibliography/documentation). The sea ice model code is freely available from the Met Office Science Repository (https://code.metoffice.gov.uk/trac/cice) under the CICE copyright agreement (https://code.metoffice.gov.uk/trac/cice/wiki/licence). The user manual for the CICE modelling code is available online (https://github.com/CICE-Consortium/CICE-svn-trunk/blob/master/cicedoc/cicedoc.pdf). JULES is available under licence free of charge. For further information on

how to gain permission to use JULES for research purposes see https://jules.jchmr.org/content/about. The code used for the integrations presented in this paper consisted of a number of branches of the GloSea code. These branches have subsequently been merged into a single package branch.

*Data availability.*  Input data files required to run the simulations described in this paper, and results from the simulations, are archived at the National Institute of Meteorological Sciences. Due to the size of the datasets used these are not routinely made available to the public; to

access GloSea5 data please contact the authors with a specific request.

*Code and data availability.*  TEXT

*Sample availability.*  TEXT

**Appendix A**

**A1**

*Author contributions.*  BWA prepared the manuscript with contributions from co-authors. BWA conceived and defined the scope of this study and performed and analysed the main assessment integrations. SML, PHC, KRK and YJK assisted with the evaluation and the interpretation of the results.

*Competing interests.*  The authors have no competing interests.

*Disclaimer.*  TEXT

*Acknowledgements.*  We acknowledge support by Korea Meteorological Administration for Research and Development for KMA Weather, Climate, and Earth system Services (153-3100-3136-304-210). We are thankful to Somin Lim (NIMS) for running and providing access to





the GloSea5 ensemble prediction data. Provision of observational data sources used within this study is also acknowledged: CryoSat2-derived thickness fields from AWI; sea ice concentration products from HadISST.



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

REFERENCE 2