# Peer review of "Seasonal sea ice forecast skills and predictability of the KMA's GloSea5"

_The Cryosphere, 2018_

## Referee Comment (RC1) · Anonymous Referee #1 · 3 Jan 2019

The paper tc-2018-217 by Byoung Woong An et al. addresses the scientific question of seasonal sea-ice forecast skill, which is relevant and within the scope of TC. Used scientific methods and assumptions seem valid and sound, and the GloSea5 system technical description in section 2.1 is rather clear. In general, the results seem sufficient to support the interpretations and conclusions presented.

However, concepts, ideas, tools and data do not appear novel. For example, it is not clear how the GloSea5 system operated by KMA differs from the developed at the UK Met Office. This shortcoming makes the description of experiments and calculations insufficiently complete and precise, and therefore do not allow their reproduction by fellow scientists. Also, if the KMA version is close to the UK Met Office one, seems strange that the UK Met Office seasonal sea-ice forecasters are not involved in this

study, and that their work is not credited and cited. In general, it seems likely that some key literature are missing in the paper as I point out next.

Importantly, the results presented in the abstract and conclusions do not appear substantial. The authors state that the sea-ice prediction was improved by implementing sea-ice thickness initial conditions and that sea-ice thickness is a key parameter for skillful prediction. But this has already been shown in many earlier studies, for example by Day et al. (2014). The specific results related to the GloSea5 system have also already been shown by Peterson et al. (2014), who are strangely is not cited in this paper, using the GloSea4 system, the predecessor of GloSea5. For example, the authors find that GloSea5 provides skillful Arctic seasonal sea-ice extent predictions up to six months and that the GloSea5 sea-ice concentration forecast skill better from October to March than from January to June. These results can also be found in Peterson et al. (2014). Moreover, the authors find that GloSea5 has a good sea-ice concentration predictability, except in summer. This finding seem to match the one by Peterson et al. (2014), who found that the GloSea4 sea-ice prediction skill for September decreases after early April due to thinning of sea ice at the start of the melt season. In summary, it is difficult to find original and important results in this paper.

Although the overall presentation follows well the general structure of a scientific paper as divided to sections, the text at the paragraph level is often very hard to read and sentence-to-sentence logic often impossible to follow, for example in Introduction. These problems arise partly because the language is not fluent and precise. Therefore the text should be inspected, rewritten and clarified. The paper is also too long (over 12 pages) for a research article in the Cryosphere and should be shortened.

Because of these shortcomings I suggest that the manuscript is rejected and recommend that the authors could submit a rewritten manuscript for review later, if substantial results are found.

References:
Day, J. J., Hawkins, E., & Tietsche, S. (2014). Will Arctic sea ice thickness initialization improve seasonal forecast skill? Geophysical Research Letters, https://doi.org/10.1002/2014GL061694

Peterson, K. A., Arribas, A., Hewitt, H. T., Keen, A. B., Lea, D. J., & McLaren, A. J. (2014). Assessing the forecast skill of Arctic sea ice extent in the GloSea4 seasonal prediction system. Climate Dynamics, 44(1–2), 147–162. https://doi.org/10.1007/s00382-014-2190-9
* * *

---

## Referee Comment (RC2) · Anonymous Referee #2 · 7 Jan 2019

This work investigates the seasonal prediction skill of Arctic sea-ice extent (SIE) and sea-ice thickness in the Korean Meteorological Administration's version of GloSea5. The authors use a variety of skill metrics to attempt to quantify the seasonal prediction skill of this system. They also claim that prediction skill is determined primarily by sea ice drift, freshwater fluxes, and sea ice thickness. The topic of this manuscript is appropriate for the Cryosphere and has potential to be of broad interest within the sea ice community.

This manuscript is generally poorly written and the scientific argumentation is difficult to follow. While having the potential to be an important study, the manuscript in present form suffers from a number of significant flaws, which I expand upon below. These flaws are sufficiently important that my recommendation for this manuscript is rejection.

[Figure]

Note: I will use the convention p.l throughout this review to refer to page number p and line number l of the discussion paper.

Major Comments:

1) The Authors claim that "Hindcast covers for the period from 1991 to 2010 and three ensemble members are initialized on fixed calendar dates, i.e., 1st, 9th, 17th and 25th." However, from what I can gather it appears that they only consider hindcasts covering the period of January 2016 to February 2017. This implies that the authors only have one, or at most two, verification dates with which to evaluate the skill of this system. It is simply not possible to evaluate a prediction system's forecast skill with so few verification dates. The authors make seemingly generic statements about the system's prediction skill throughout the manuscript (and in the title), however these statements are invalid given the extremely short hindcast period considered. The authors must either (i) expand the number of hindcast years so that statistically robust statements about prediction skill can be made; or (ii) recast this work as a seasonal prediction case study focusing on the year 2016. If they opt for (ii) they should strictly avoid using the term "forecast skill" in the manuscript.

2) Given that the results presented in this manuscript do not represent a robust prediction skill assessment, these findings add little to the existing literature on seasonal Arctic sea ice prediction. The authors rightfully mention a number of studies that have used hindcasts to examine prediction skill (each based on a set of hindcast experiments spanning at least 20 years), but make little to no effort to place their results in the context of these earlier works.

3) Many aspects of the methodology and the results of the manuscript are unclear, including:

3.17: It is unclear what initialization dates, what ensemble sizes, and what time period are used for these hindcasts.

[Figure]

5.5: It is unclear what is being shown in Fig. 1. Is this the ensemble spread from forecasts initialized on Jan 1, 2016? The forecasts only run for 240 days, so are multiple initialization dates shown in this figure? This is not clear from the text or caption.

5.17: As with Figure 1, it is unclear what is being plotted in Fig. 2. What initialization dates are being used here? Also, what units are used in Fig. 2b? RMSE should be in kmˆ2.

8.3: Ice that is thicker than 30cm is commonly seen near the Siberian coastline in winter, which contradicts the authors' statement here.

8.13: Where is this snow depth verification data coming from? My understanding is that CryoSat-2 uses the Warren climatology for snow depth, which by definition does not have any interannual variability.

11.3: It is unclear how this correlation is being computed. There is only one verification period available for each of these forecasts. How are the authors defining a correlation here? What quantities are being correlated?

11.7: What SIT data is being used for the months of May and June (no CryoSat-2 data is available in this period)?

13.5: Why is the CRPSS so drastically different than the correlation and generalized discrimination scores?

14.23: It is unclear what figure in the manuscript supports this statement.

14.28: This statement is unsupported by the results of the manuscript.

15.2: It is unclear what this statement means.

15.5: It is unclear how the manuscript has shown this.

15.6-7: This has also not been shown.

15.18: The results do not show this.
4) The manuscript makes no connection with previous work on sea-ice prediction performed with the GloSea system. In particular, Peterson et al. (2015) assess Arctic sea ice prediction skill using the GloSea4 system. Comparisons should be made to the model biases and forecast skill results of Peterson et al (2015). Also, information should be provided on the key differences (in terms of both initialization and model formulation) between GloSea4 and GloSea5. The authors should attempt to relate any skill differences to the differences between these systems.

References:

Peterson KA, Arribas A, Hewitt H, Keen A, Lea D, McLaren A (2015) Assessing the forecast skill of Arctic sea ice extent in the GloSea4 seasonal prediction system. Clim Dyn 44(1–2): 147–162

---

## Author Comment (AC1) · 1 Mar 2019

Journal: TC Title: Seasonal sea ice forecast skills and predictability of the KMA's GloSea5 Author(s): Byoung Woong An et al. MS No.: tc-2018-217 MS Type: Research article

We would like to thank the reviewers for careful and thorough reading of this manuscript and for the thoughtful comments and constructive suggestions, which help to improve the quality of this manuscript. Our response follows.

Response to RC1: Comments from Referee: However, concepts, ideas, tools and data do not appear novel. For example, it is not clear how the GloSea5 system operated by KMA differs from the developed at the UK Met Office. This shortcoming makes the

description of experiments and calculations insufficiently complete and precise, and therefore do not allow their reproduction by fellow scientists. Also, if the KMA version is close to the UK Met Office one, seems strange that the UK Met Office seasonal sea-ice forecasters are not involved in this study, and that their work is not credited and cited. In general, it seems likely that some key literature are missing in the paper as I point out next. Importantly, the results presented in the abstract and conclusions do not appear substantial. The authors state that the sea-ice prediction was improved by implementing sea-ice thickness initial conditions and that sea-ice thickness is a key parameter for skillful prediction. But this has already been shown in many earlier studies, for example by Day et al. (2014). The specific results related to the GloSea5 system have also already been shown by Peterson et al. (2014), who are strangely is not cited in this paper, using the GloSea4 system, the predecessor of GloSea5. For example, the authors find that GloSea5 provides skillful Arctic seasonal sea-ice extent predictions up to six months and that the GloSea5 sea-ice concentration forecast skill better from October to March than from January to June. These results can also be found in Peterson et al. (2014). Moreover, the authors find that GloSea5 has a good sea-ice concentration predictability, except in summer. This finding seem to match the one by Peterson et al. (2014), who found that the GloSea4 sea-ice prediction skill for September decreases after early April due to thinning of sea ice at the start of the melt season. In summary, it is difficult to find original and important results in this paper. Although the overall presentation follows well the general structure of a scientific paper as divided to sections, the text at the paragraph level is often very hard to read and sentence-to-sentence logic often impossible to follow, for example in Introduction. These problems arise partly because the language is not fluent and precise. Therefore the text should be inspected, rewritten and clarified. The paper is also too long (over 12 pages) for a research article in the Cryosphere and should be shortened. Because of these shortcomings I suggest that the manuscript is rejected and recommend that the authors could submit a rewritten manuscript for review later, if substantial results are found.

[Figure]

Authors' response: We thank the reviewer for the detailed and helpful comments. We have now addressed those comments by adding a new paragraph in which we lay out our findings.

Authors' changes in manuscript: Abstract: Accurate prediction of the Arctic summer sea ice remains crucial in most numerical models for the global seasonal forecast. Ensemble seasonal sea ice forecasts of the Arctic Ocean conducted with the coupled global seasonal forecast system (GloSea5) is verified. Here we examine the verification measures for assessing the overall performance of ensemble forecasts as probabilistic forecasts. To investigate the temporal and spatial accuracy of the seasonal prediction of the Arctic sea ice extent and thickness of the year 2016, a set of ensemble probability skill metrics including anomaly correlation, CRPSS, and the generalized discrimination score are assessed. We used the forecast data up to six months for the validation of the model predictability which is based on the anticipated condition for the lead-month span. It shows significance for all lead months except for anomalous errors around East Siberian Sea, Chukchi Sea and Beaufort Sea during summer months. Considering the forecast skill, we found the sea ice forecast accuracy is determined primarily by the timing of sea ice drift (i.e., Beaufort Gyre and Transpolar drift) and sea ice formation by freshwater flux in the East Siberian Sea.

Authors' response: Regarding the length of the article, it is not easy to shorten the article. However, as suggested by the reviewer, we have reviewed carefully the entire manuscript and have removed irrelevant information as shown in the revised manuscript.

Response to RC2:

Comments from Referee: The Authors claim that "Hindcast covers for the period from 1991 to 2010 and three ensemble members are initialized on fixed calendar dates, i.e., 1st, 9th, 17th and 25th." However, from what I can gather it appears that they only consider hindcasts covering the period of January 2016 to February 2017. This

implies that the authors only have one, or at most two, verification dates with which to evaluate the skill of this system. It is simply not possible to evaluate a prediction system's forecast skill with so few verification dates. The authors make seemingly generic statements about the system's prediction skill throughout the manuscript (and in the title), however these statements are invalid given the extremely short hindcast period considered. The authors must either (i) expand the number of hindcast years so that statistically robust statements about prediction skill can be made; or (ii) recast this work as a seasonal prediction case study focusing on the year 2016. If they opt for (ii) they should strictly avoid using the term "forecast skill" in the manuscript.

Authors' response: This is a case study focused on prediction accuracy of the year 2016 of GloSea5 ensemble forecast operated by KMA, so we didn't consider the historical re-forecast data for the calculation of the forecast skill at all. We agree with the reviewer's advice and have therefore revised the title for clarity and the recommended term as "A seasonal sea ice prediction case study focusing on the year 2016 using GloSea5 operated by KMA" and replaced the "forecast skill" to "prediction accuracy" within the manuscript.

Given that the results presented in this manuscript do not represent a robust prediction skill assessment, these findings add little to the existing literature on seasonal Arctic sea ice prediction. The authors rightfully mention a number of studies that have used hindcasts to examine prediction skill (each based on a set of hindcast experiments spanning at least 20 years), but make little to no effort to place their results in the context of these earlier works.

Authors' response: We agree with the reviewer's advice and have therefore we added the earlier works related to the Arctic sea ice forecast skills in the manuscript. Authors' changes in manuscript: There have been some studies on potential for sea ice predictions by a set of studies (e.g. Kauker et al., 2009; Koenigk and Mikolajewicz, 2009; Holland et al., 2010; Day et al., 2014). There have also been studies on operational sea ice forecast skill predicted from ocean-sea ice model forecasts (Zhang et al.,

2008; Lindsay et al., 2012), and from fully-coupled general circulation models (GCMs) (Wang et al., 2013; Chevallier et al., 2013; Sigmond et al., 2013). The role of the initial ice thickness distribution is emphasized for the forecast quality in previous studies (Kauker et al., 2009; Holland and Stroeve, 2011; Lindsay et al., 2012; Chevallier and Salas-Mélia, 2012). To improve the initial state estimation, it is suggested to use observational information in a data assimilation system systematically (Lindsay et al., 2012; Chevallier et al., 2013; Yang et al., 2014; Massonnet et al., 2015). Consequently, a dynamical forecast system that seeks to predict summer Arctic sea ice conditions should rely on realistic initial conditions. In particular, the predictability of summer Arctic sea ice is known to reside in its thickness (Holland and Stroeve, 2011; Chevallier and Salas-Mélia, 2012; Wang et al., 2012). However, few sea ice thickness measurements are available for direct assimilation because observations of sea ice thickness are sparse and inaccurate (Mathiot et al., 2012; Lindsay et al., 2012). In addition to the importance of the initial state, seasonal Arctic sea ice prediction by dynamical models need a good knowledge of boundary conditions of the simulation period (Kauker et al., 2009). The seasonal Arctic sea ice forecast by climate models is to identify the future occurrence of anomalies from the long-term average. The Met Office seasonal prediction system known as GloSea is one of only a few operational seasonal prediction systems including the initialization of observed sea ice followed by its prognostic determination in a coupled dynamical model of sea ice. Peterson et al. (2015) assessed the ability of GloSea4 (Arribas et al., 2011) seasonal forecast skill and showed the best forecast skill of the September monthly mean ice extent when the system was initialized in late March and early April, as determined from the historical forecast period of 1996–2009 with correlation skills in the range of 0.6. However, correlation skills using May initialization dates were much lower due to thinning of the sea ice at the start of the melt season which allows ice to melt too rapidly. Currently GloSea5 operated by UK Met Office is determined from the period of 1993-2015, while GloSea5 operated by KMA is from the historical forecast period of 1991-2010. As described above, the sea ice extent predictability is partly sensitive to initialization and atmospheric conditions.

[Figure]

These two GloSea5 operators use their own initial state and atmospheric boundary condition over the forecasting period as input. Therefore forecasts from each operator could be slightly different in the sea ice extent predictability.

Many aspects of the methodology and the results of the manuscript are unclear, including: 3.17: It is unclear what initialization dates, what ensemble sizes, and what time period are used for these hindcasts. Authors' response: Every day we perform four ensemble members' initialization with 0000 UTC analyses from the NWP global data assimilation and the ocean–sea‐ice data assimilation system. Two of these members were executed for 210 days each (seasonal forecast members) and the other two were executed for 60 days each (intraseasonal members). Seasonal forecast members from the previous three weeks are combined, resulting in a 42‐member ensemble for the next six months. These products are updated on a weekly basis. Generating the forecast on a weekly cycle means that frequent updates can be given when required. Each individual simulation, referred to hereafter as an ensemble member, is performed with the same model version and with the same external forcing. In this study, we evaluate the prediction accuracy during the year 2016 forecast data with global coupled model, GloSea5. Its seasonal set-up has been performed over the Arctic region daily. The model evaluation is focused on monthly ensemble means of sea-ice concentration, sea-ice thickness and snow-depth.

5.5: It is unclear what is being shown in Fig. 1. Is this the ensemble spread from forecasts initialized on Jan 1, 2016? The forecasts only run for 240 days, so are multiple initialization dates shown in this figure? This is not clear from the text or caption. Authors' response: In general the spread generated by an ensemble forecast is a good indication of its error. Therefore we examine how ensemble members diverge over time assessed by calculating the average error of all forecasts with an ensemble spread within a certain range. Shorter-range forecasts generally have a large number of start dates, while seasonal forecasts have significantly less. Therefore the dates indicated in Figures 1 and 2 represent a fixed date as one per month to carry out forecast verification. Authors' changes in manuscript: Figure 1 caption modified: Ensemble spread error range of monthly Arctic sea ice extent forecast with 42 ensembles from January 2016 to February 2017. Ranges are monthly averaged sea ice extent errors from each ensemble member of GloSea5 forecast by KMA from January 2016. The unit of the error is ãĂŰ10ãĂŮ̂6 km̂2. In each boxplot, the bottom, middle and top of the box represent 25th, 50th and 75th percentiles respectively, and the bottom and top whiskers represent 10th and 90th percentiles respectively.

5.17: As with Figure 1, it is unclear what is being plotted in Fig. 2. What initialization dates are being used here? Also, what units are used in Fig. 2b? RMSE should be in kmЁĘ2. Authors' response: We tried to clarify what has been unclear in the previous manuscript regarding Figure 1 and 2. Authors' changes in manuscript: Figure 2 caption modified: HadISST's value subtracted Ensemble mean bias (a), Root Mean Square Error (b) and Nash-Sutcliffe Index (c). The unit of the RMSE is ãĂŰ10ãĂŮ̂6 km̂2. In each boxplot, the bottom, middle and top of the box represent 25th, 50th and 75th percentiles respectively, and the bottom and top whiskers represent 10th and 90th percentiles respectively.

8.3: Ice that is thicker than 30cm is commonly seen near the Siberian coastline in winter, which contradicts the authors' statement here.

Authors' response: The text is indeed incorrect. The sentence has been corrected to read. Authors' changes in manuscript: The presence of thick sea ice along the coast of East Siberian in October is also different from that of observed.

8.13: Where is this snow depth verification data coming from? My understanding is that CryoSat-2 uses the Warren climatology for snow depth, which by definition does not have any interannual variability. Authors' response: This operational prediction systems show some skills in predicting winter snow depth, but diagnosing the source of forecast errors is problematic. Such forecast errors may be due to incomplete knowledge of the initial state of key variables such as sea ice thickness which is not well

observed (Mathiot et al., 2012).

11.3: It is unclear how this correlation is being computed. There is only one verification period available for each of these forecasts. How are the authors defining a correlation here? What quantities are being correlated? Authors' response: We calculated the Anomaly Correlation Coefficient with the ensemble mean of the model forecast and the corresponding reference data for each start date and each lead month. Sea ice concentration and sea ice thickness are being calculated in correlations with HadISST and CryoSat-2 data, respectively.

11.7: What SIT data is being used for the months of May and June (no CryoSat-2 data is available in this period)? Authors' response: There is no available CryoSat-2 SIT data for May and June and we only calculated from January to April, 2016. We have now corrected SIT JFMAMJ to SIT JFMA and changed in the manuscript.

13.5: Why is the CRPSS so drastically different than the correlation and generalized discrimination scores? Authors' response: To examine the performance of ensemble forecasts as probabilistic forecasts of a continuous variable, we evaluated the forecast quality of a set of ensemble forecasts as a continuous function of the forecast variable. Rather than using a single value to describe some aspect of forecast quality, combining measures of correlation and discrimination is needed to completely describe the prediction skill. This is helpful for diagnosing how these aspects affect forecast accuracy. Summarizing the results of all three the correlation, discrimination and CRPSS suggest that ensemble forecasts may contain probability forecast statements that are of high quality (skillful) for predicting sea ice thickness in the Beaufort Gyre and the Transpolar drift, but are of low quality (no skill) for predicting sea ice concentration over Arctic Ocean.

14.23: It is unclear what figure in the manuscript supports this statement. Authors' changes in manuscript: The September outlook from the June initialization is relatively unbiased. However, large biases are developed when initialized in March (see Figure

3).

14.28: This statement is unsupported by the results of the manuscript. Authors' response: As the result of the ensemble prediction examination, using the 75th percentile of the Ensemble value would provide excellent forecasting for other seasons.

15.2: It is unclear what this statement means. Authors' changes in manuscript: The correlation analysis between the forecast and observed sea ice show that the predictive skill is associated with all the lead times except for the summer season. This is because the model is capable of predicting the Arctic sea ice, but there is no obvious correlation between the extent and lead time for the summer sea ice prediction.

15.5: It is unclear how the manuscript has shown this. Authors' response: Assessing the divergence among the ensemble members reveal that sea ice exhibits operational predictability for winter conditions, but shows little predictability for the spring season.

15.6-7: This has also not been shown. Authors' response: As shown in Figures 6∼8, the operational predictability of sea ice forecasts indicates that sea ice concentration in higher variable sea ice regimes generally show higher predictability for a longer period of time.

15.18: The results do not show this. Authors' response: Since the model set up was for seasonal scale simulation, we examined the skill of the capturing the seasonal sea ice variability of observations. Figure 3 shows a reasonably good prediction of the sea ice extent performed by the model. The model is also able to capture the summer sea ice melting but very little. Hence potentially, this model is useful in seasonal prediction of sea ice extent for Arctic.

4) The manuscript makes no connection with previous work on sea-ice prediction performed with the GloSea system. In particular, Peterson et al. (2015) assess Arctic sea ice prediction skill using the GloSea4 system. Comparisons should be made to the model biases and forecast skill results of Peterson et al (2015). Also, information

should be provided on the key differences (in terms of both initialization and model formulation) between GloSea4 and GloSea5. The authors should attempt to relate any skill differences to the differences between these systems. Authors' response: We omitted the previous study (i.e., Peterson et al., 2015), due to the differences in system, methodology and objective of the study. In addition, we focused on sea ice prediction accuracy from statistical analysis and applied this to dynamical interpretation on sea ice variability. Therefore we could not compare our data with Peterson et al.'s data. However, we recognized that it is closely related to this study, so we have included this reference in our manuscript. Authors' changes in manuscript: The main difference between GloSea4 and GloSea5 system is horizontal resolutions both in the atmosphere and the ocean. GloSea4 has a horizontal resolution of approximately 120 km at mid-latitudes (N96) in the atmosphere, and nominally 1 ° horizontal resolution in the ocean (Madec, 2008). The major upgrade to GloSea5 was implemented to increase the horizontal resolution to 50 km in the atmosphere (N216) and 0.25 ° in the ocean (MacLachlan et al. 2014), with the ocean and sea ice analysis also being upgraded to a three dimensional variational system (FOAMV12) (Blockley et al. 2014). GloSea5 (GloSea4) operated by UK Met Office was determined from the period of 1993-2015 (1996-2009), whereas GloSea5 operated by KMA was from the historical forecast period of 1991-2010.
* * *